# Enhancement of the Solubility of BS Class II Drugs with MOF and MOF/GO Composite Materials: Case Studies of Felodipine, Ketoprofen and Ibuprofen

**DOI:** 10.3390/ma16041554

**Published:** 2023-02-13

**Authors:** Jinyang Han, Bo Xiao, Phung Kim Le, Chirangano Mangwandi

**Affiliations:** 1School of Chemistry and Chemical Engineering, Queen’s University Belfast, David Keir Building, Stranmillis Road, Belfast BT9 5AG, UK; 2Faculty of Chemical Engineering, Ho Chi Minh City University of Technology (HCMUT), 268 Ly Thuong Kiet Street, District 10, Ho Chi Minh City 70000, Vietnam; 3Vietnam National University Ho Chi Minh City, Linh Trung Ward, Thu Duc City, Ho Chi Minh City 70000, Vietnam

**Keywords:** HKUST-1, graphite oxide, Class II drugs, felodipine, ketoprofen, ibuprofen

## Abstract

In this research, a novel composite material composed of Metal-Organic Framework material (MOF) and graphite oxide was synthesized and evaluated as a possible drug-loading vehicle. HKUST-1, a MOF material originally designed by the Hong Kong University of Science and Technology, was used as a model porous material. The aim was to synthesize a drug delivery vehicle for modifying the release kinetics and solubility of poorly soluble drugs (BSC Class II drugs); these are drugs that are known to have poor bioavailability due to their low solubility. We used ketoprofen, ibuprofen, and felodipine as models for BSC Class II drugs. The drugs were loaded onto composite materials through adsorption. The adsorption of these three drugs into the matrix of HKUST-1/GO (graphite oxide), HKUST-1, and graphite oxide was compared. The loading efficiency of the drugs onto the carrier was dependent on the drug molecule and the composition of the drug carrier. The inclusion of graphite oxide in the drug carrier matrix improved the drug loading capacity and modified the drug release rate. The loading of the three drugs felodipine, ketoprofen, and ibuprofen onto HKUST-1 were 33.7, 58, and 79 mg/g respectively. The incorporation of GO into the HKUST-1 matrix resulted in an increase in the loading by 16 and 4 mg/g for the ketoprofen and ibuprofen drugs. When compared to the pure drugs, the solubility of all three drugs in the HKUST-1/GO matrix increased by at least 6 folds.

## 1. Introduction

The solubility of drug molecules is an imperative factor that needs to be considered during the development of new drug products. The current drug selection procedures, such as combinatorial chemistry and high throughput screening, allow potentially effective drug candidates to be easily chosen as the target based on their high affinity and selectivity. According to the Biopharmaceutical Classification System (BCS), Class II drugs are characterized by poor solubility and high permeability in the human body [1]. It has been reported that the main obstacle encountered by more than 50 percent of newly developed chemical entities from drug discovery programs is their low solubility in water. Improving the oral bioavailability of poorly water-soluble drugs by increasing the dissolution rate is one of the main challenges faced in drug development [2].

Several approaches have been used to improve the water—solubility and hence increase the bioavailability of BSC class II drugs, for example, particle size reduction, salt formation, solubilization, lipid–based drug delivery, and solid dispersion (SD). A solid dispersion is a distribution of one or more active ingredients in an inert carrier or matrix where the active ingredient could exist in the finely crystalline, solubilized, or amorphous state [3]. Theoretically, in the SD system, the active pharmaceutical ingredients can be dispersed in a solid matrix carrier, either as a eutectic or a phase-separated mixture, or as an amorphous solid dispersion (ASD) [3].

Another approach is to load poorly soluble drugs into a porous medium, which then aids drug distribution when the medium disintegrates. Several materials have been tested for this, such as metal oxides, porous silica [4], zeolites [5,6], and more recently metal-organic frameworks (MOF). Metal-organic frameworks (MOFs), a kind of porous hybrid inorganic-organic solids, have recently attracted great interest in many research areas such as catalysis [7], separation [8], gas adsorption and storage [9,10], and also drug delivery [11,12,13,14,15]. MOFs have also found applications in other areas such as the separation of organic compounds [16]; removal of pollutants from solutions [17,18]; degradation of volatile organic compounds [19]; production of materials for energy storage devices [20,21]; design of sensors [22,23] and also in cancer therapy [24,25].

MOFs have advantageous characteristics such as adjustable pore size, high surface area, and diverse functionalities. They could be intrinsically biodegradable if their relatively labile metal-ligand bonds are decomposed in the biological environment [12]. The size of the pores can be tuned by carefully selecting organic ligands and metal cations as the building blocks [26]. These characteristics of MOFs make them attractive candidates as drug delivery vehicles. There are reports in the literature on the application of MOFs as the carrier for BSC class II drugs [6,11,12,27] Despite the success in the use of these MOFs as drug delivery vehicles, there are still some challenges. For instance, the size of the MOF cages can restrict the choices of drugs. There are also biocompatibility issues as some MOFs consist of toxic metal ions. Some are not stable under aqueous conditions. For instance, the Hong Kong University of Science and Technology (HKUST-1) metal-organic framework is known to have poor stability in water solutions.

Researchers have explored the use of composites made from MOFs and other materials to improve biocompatibility and stability under aqueous conditions. Graphene oxide (GO) is known to be biocompatible and stable in aqueous media and has a large surface area. Therefore, MOF/GO composite systems have been tested as vehicles of gas storage, adsorbents, and also as catalysts [28,29,30,31].

Attempts to use MOF composites as drug delivery vehicles for controlling drug release have also been reported in the literature [4,32,33]. Earlier work reported by Sun and others showed that a composite derived from MOF HKUST-1 and GO can improve the loading capacity of ibuprofen, the BCS Class II drug [4]. A further systematic study is still required to understand this new drug delivery system for the storage and release of BSC Class II drugs. In this research, GO, MOF HKUST-1, and composite HKUST-1-GO were investigated for loading three typical BCS Class II drugs, namely felodipine, ibuprofen, and Ketoprofen, which includes studies of drug loading and releasing kinetics to understand the effect of the structural parameters of MOF/GO on BCS Class II drug delivery.

## 2. Materials and Methods

Methanol (CH_3_OH), phosphoric acid (H_3_PO_4_), hydrochloric acid (HCl), sulphuric acid (H_2_SO_4_), graphite, potassium permanganate (KMnO_4_) copper nitrate, hydrogen peroxide (H_2_O_2_) and benzene -1-3-tricarboxylic acid (BTC) were all purchased from Sigma Aldrich, Bellingham, UK. Felodipine was manufactured by AstraZeneca and supplied by Sigma Aldrich, UK; ketoprofen and ibuprofen with purity >98% were supplied by Sigma Aldrich. The chemical structures of these drugs are shown in Figure 1.

### 2.1. Preparation of Materials

#### 2.1.1. A HKUST-1 Synthesis

The HKUST-1 was synthesized using a hydrothermal method [32]. The procedure involves making a 10 mL solvent solution consisting of methanol and deionized water (in a ratio of 1:1 *v*/*v*). After preparing the solution, 6.6 mmol of BTC was then added to the solvent followed by 10.12 mmol of copper nitrate, making sure that both dissolved by stirring. The mixture with then transferred into an autoclave and heated at 110 °C for 24 h. At the end of the reaction, the precipitate formed was separated by filtration and dried before further characterization tests, which included Fourier Transform Infrared Spectroscopy (FTIR), using Perkin Spectrum 100 (Perking Elmer, Waltham, MA, USA), Brunauer–Emmett–Teller (BET), and Scanning electron microscopy (SEM) was determined using JSM 6400 scanning micro-scope (Jeol, Peabody, MA, USA). X-ray diffraction (XRD) spectra of the samples was determined using Philips PANalytical X’pert pro diffractometer (manufactured by PANanalytical, Westborough, MA, USA).

#### 2.1.2. GO Synthesis

Graphene oxide (GO) was produced using the modified Hummers method [20] from pure graphite powder; in this method, 54 mL of sulphuric acid (H_2_SO_4_) and 6 mL of phosphoric acid (H_3_PO_4_) (volume ratio 9:1) were mixed and stirred for several minutes. Then 0.45 g of graphite powder was added into the mixing solution under stirring conditions. 2.64 g of potassium permanganate (KMnO_4_) was then added slowly into the solution. This mixture was stirred for 10 h until the solution became dark green. To eliminate the excess KMnO_4_, 1.35 mL of hydrogen peroxide (H_2_O_2_) was dropped slowly and stirred for 10 min. The exothermic reaction occurred and the mixture was allowed to cool down to room temperature. 10 mL of hydrochloric acid (HCl) and 30 mL of deionized water (DIW) were added and centrifuged using Eppendorf Centrifuge 5430R (supplied by VWR, Leicestershire, UK) at 5000 rpm for 30 min. Then, the supernatant was decanted away, and the residuals were then rewashed three times with HCl and DIW. The washed GO solid was dried using an oven at 90 °C for 24 h to produce the powder of GO. The GO was characterized using FTIR, SEM, and XRD techniques.

### 2.2. Synthesis of MOFs-Composite

The HKUST-1/GO composite was synthesized by dispersing GO powder in the well-dissolved BTC. The resulting suspensions were subsequently stirred and subjected to the same synthesis procedure as for HKUST-1. The added GO consisted of 5, 10, and 15 wt% of the final material.

### 2.3. Characterization of the Samples -FTIR, SEM XRD BET

XRD analysis was performed to examine the crystalline nature of the samples (wide-angle X-ray diffraction) using a Philips PANalytical X’pert pro diffractometer (manufactured by PANanalytical, USA). The structural ordering of the samples was analyzed by scanning electron microscopic (SEM), using a JEOL-JSM 6400 scanning microscope (Jeol, USA). The surface area of the samples was calculated to form N_2_ adsorption measurements performed at a temperature of 77 K, using the Brunauer, Emmett, and Teller (BET) equation. The samples were analyzed before and after drug loading.

### 2.4. Drug Loading Tests

Before loading, the MOFs/MOFs composite was activated. Class II drugs (Felodipine, ibuprofen, ketoprofen) were first dissolved in the solvent (50 mL), and then GO/HKUST-1/HKUST-1-composite (500 mg) was added. Cyclohexane was used as the solvent for making the solutions for the loading of ibuprofen and ketoprofen. Felodipine was dissolved in acetone to make solutions for drug-loading tests. After sealing the vile tightly, the mixture was allowed to stand for 72 h at room temperature and the supernatant was analyzed by UV-VIS spectroscopy. The amount of drug loaded was determined using:(1)$$q_{e}=\frac{c_{o}-c_{e}}{m}V_{d}$$

In Equation (1), $c_{o}$ is the initial drug concentration in solution, $c_{e}$ is the concentration at equilibrium, $V_{d}$ is the volume of the drug solution and m is the mass of the drug-loading material added.

At the end of the loading process, MOF/MOF-GO composite materials were separated by filtration and dried in a vacuum oven, and kept in a desiccator before further analysis.

In the kinetics, study samples were withdrawn from the reactor vessel at pre-determined times. Ultraviolet-visible spectroscopy (UV) analysis was performed on the samples to determine the concentration of drug remaining in solution, $c_{t}$. The amount of drug loaded at time, t, was calculated using:(2)$$q_{t}=\frac{c_{o}-c_{t}}{m}V_{d}$$

In Equation (2), $c_{t}$ is the concentration of drug solution at time *t* and all the other symbols are as previously defined.

The kinetics of the drug loading was described by the following equation:(3)$$q_{t}=q_{m}\left(1-exp\left(-k_{L}t\right)\right)$$

$q_{t}$ is the drug loading after a time *t*, $k_{L}$ is a parameter representing the loading rate; $q_{m}$ is the maximum loading capacity achieved. 

### 2.5. Drug Release

The MOF and MOF-GO materials (0.5 g) that had been previously loaded with the drugs were dispersed in 50 mL of phosphate buffer solution at 37 °C. The solutions were agitated with a mechanical stirrer (800 rpm). Samples were drawn at pre-determined times and the amount of drug released into phosphate buffer solution (PBS) was analyzed using UV-vis spectroscopy. The cumulative amount of drug released was calculated from:(4)$$m_{d,\;released}=\overset{n}{\underset{i=1}{\sum }}c_{i}V$$

We assumed that during the drug-loading step, the drug was uniformly distributed in the carrier material such that results obtained from loading tests were representative of any sample taken from the carrier material. Therefore, the mass of the carrier material used in the dissolution/release test is the amount of drug available for release given by:(5)$$m_{d,\;available}=q_{e}m_{s}$$

The cumulative fraction of drug release is given by:(6)$$\eta =\frac{m_{d,released}}{m_{d,available}}=\frac{\sum_{i=1}^{n}c_{i}V}{q_{e}m_{s}}$$
where $q_{e}$ and V are the drug loading achieved (mg/g) and volume of dissolution medium, $c_{i}$ is the concentration of the i^th sample collected at a time, *t*.

The fraction of drug released was described by the following equation:(7)$$\eta =\left(1-\exp\left(-k_{R}t\right)\right)$$
where *η* is the drug loading after a time *t*, $k_{R}$ is a parameter representing the drug release rate.

## 3. Results and Discussion

### 3.1. Sample Characterisation

Figure 2a shows the powder X-ray diffraction results of the HKUST-1 and the MOF—Graphite composite materials. All of the diffraction peaks of HKUST-1 were in line with those reported in the reference [34], and the indexed values of all of the diffraction peaks were in accordance with the literature [35]. From the information presented in Figure 2, the diffraction patterns of the HKUST-1 with 5%GO, 10%GO, and 15%GO are almost the same. Also, the HKUST-1/GO composite is similar to that observed for HKUST-1, which indicates that the structure of HKUST-1 is preserved even after the addition of GO. The peak around 10° is slightly reduced in the composite, which suggests that the addition of GO is affected by the HKUST-1 in the composite. No significant differences were observed in the XRD spectra of the HKUST-1 and HKUST-1/Go composites before and after drug loading.

Further confirmation of HKUST-1/GO is provided by the FT-IR spectra from Figure 2. The spectra of HKUST-1 and that of the composite look rather similar. All the characteristic peaks are also observed in HKUST-1/GO, which shows that the structure of HKUST-1 is preserved. The characteristic peaks at 938 cm^−1^ and 1640 cm^−1^ of GO are both observed in HKUST-1/GO. This indicates that the structure of GO is also preserved in the HKUST-1/GO composite. The characteristic peaks of GO were observed at 1725 cm^−1^, 1618 cm^−1^, 1388 cm^−1^, and 1031 cm^−1^, which is in close agreement with the literature [36,37]. In the GO spectra the peak at 1031 cm^−1^ is ascribed to the functional group C-O and C=O is attributed to the peak at 1388 cm^−1^. The peak at around 3427 cm^−1^ is ascribed to the hydroxyl groups. The asymmetric vibrations of the carboxylate groups are attributed to peaks at 1645 cm^−1^, in which the symmetric vibrations of the same functional groups are ascribed to 1376 cm^−1^. The peaks detected in the range 700 to 1700 cm^−1^ confirmed the presence of the benzene tricarboxylic acid (BTC) as the organic ligand in the structure. The two other functional groups related to HKUST-1 and HKUST-1/GO are observed at around 730 cm^−1^ and 1108 cm^−1^ which correspond to C-H and C-O bonds respectively. Similarly, no significant differences were observed between FTIR samples of materials before and after drug loading. This could be due to the low levels of drug loading in the samples.

The SEM microphotographs were used to investigate the morphologies, crystallinity, and microstructures of HKUST-1/MOF and MOFs composite. As shown in Figure 3. It has been confirmed from the SEM that there are no other phases present except the blocks of the produced MOF. Thus, it can be concluded that the synthesized samples were free from unreacted chemicals. GO is seen as dense flakes of graphene sheets stacked together, whereas HKUST-1 is in a crystalline phase similar to the observations of Sun et al. [4]. The structure of HKUST-1/GO composites is very different from that of HKUST-1 and GO pristine materials. It consists of regular thin platelets stacked together in an organized way. This shows that a new kind of composite material has been produced that is not just a simple physical mixture of HKUST-1 and GO.

The results for BET surface area are presented in Table 1. The BET surface area data for HKUST obtained here are similar to other reports from the literature [5,30,31,32,33,38,39,40,41]; Hartmann et al. 2008 reported a BET surface area ranging between 1153 and 1624 m^2^/g for samples produced under different ranges of conditions [39].

It could be seen from the results that the addition of GO has increased the surface area of the composite material, which agrees with results from other reports in the literature [4,28,29]. Work by Mohammadabad et al. 2020 showed that the addition of 2.5% of GO to HKUST-1 results in about a 30% increase in BET surface area [42]. It would therefore be anticipated that the performance of MOF composite material will be improved.

### 3.2. Drug Loading

#### 3.2.1. Loading on Graphite Oxide and HKUST-1 Matrices

Figure 4 shows the typical UV spectra of samples drawn at different times during the loading of Ibuprofen onto the HKUST-1 composite material. As expected, the intensity of the characteristic peaks decreased with increasing contact time, indicating a decrease in the drug concentration in the solution. This implies the successful loading of the drugs on the solid phase.

Figure 5 shows the results of the drug loading of three BCS Class II drugs onto graphene oxide (GO). Figure 5a shows of the fraction drug remaining in the solution as a function of time. Figure 5b shows the amount of the drug-loaded onto the solid phase as a function of time. It is clear from Figure 5 that the loading capacity of GO for the different drugs is different drugs. However, the common observation here is that maximum loading is achieved between 12 and 24 h of contact time. Different levels of loading were achieved for the different drugs, and loading increased in the order felodipine < ketoprofen < ibuprofen.

For felodipine, the maximum loading achieved was only 1.2% of the available drug in the solution. A slightly higher loading (~6%) was achieved for ketoprofen. The highest loading was achieved for ibuprofen (~9%). The difference in the levels of drug loading could be due to the difference in the loading mechanisms due to the differences in the molecular structure of the drugs. Ketoprofen and ibuprofen both have carboxyl groups in their structures (see Figure 1). For ibuprofen and ketoprofen drugs, the possible loading mechanism is chemisorption, which involves coordination bonding between the carboxyl group in drugs and the hydroxyl and carboxyl groups in the GO [4]. The loading of felodipine on GO could be mainly due to physisorption, which is governed by van der Waals forces between the drug molecules and active sites on the GO. The initial loading rates of the three drugs are also different.

Figure 6 illustrates the loading of the three drugs on the HKUST-1 matrices. In comparison to GO, HKUST-1 has better performance in terms of drug loading for all the 3 drugs. For example, after 72 h of contact, 32.5% of the felodipine drug was loaded onto HKUST-1 compared to a mere 1.2% loading onto GO. Ibuprofen loading onto HKUST-1 was about 77.2% which is significantly higher than it was on GO (~9.2%). Similar to GO material, different loading efficiencies were observed for the different drugs, and the order in which the loading efficiency increased is the same as observed for HKUST-1. The difference in loading capacities of GO and HKUST-1 may be attributed to the difference in the available surface area and the average size of the pores. The surface area of HKUST-1 is more than twice that available in GO (see Table 1).

#### 3.2.2. Loading on HKUST-1/GO Composites

The drug loading performance of composites with different amounts of GO ranging from 5 to 10% (*w*/*w*) is shown in Figure 7. The effect of increasing the composition of GO from 5 to 10% on loading of felodipine to HKUST-1/GO composite is presented in Figure 7b while Figure 7b,c shows results for ketoprofen and ibuprofen respectively. Figure 7d shows a comparison of the loading of the 3 drugs at this highest loading of GO.

The continuous lines in the plots presented in Figure 7 are non-linear regression fits of Equation (3) to the drug loading profiles. The drug-loading parameters of the different drugs on the different drug-loading vehicles are summarized in Table 2.

For all three drugs investigated, the inclusion of GO into the formulation resulted in increased loading of the drug. It can be noticed from the data in Table 1 that loading capacities for the formulations with 10 and 15% GO are not significantly different for all the drugs. This implies that increasing the composition of GO beyond 10% does not provide any additional benefits.

### 3.3. Drug Release

Figure 8a–c shows the release profiles of felodipine, ketoprofen, and ibuprofen respectively. It could be seen that the felodipine release from MOFs and MOFs composite are very soon and quite similar, that might be because the felodipine is weakly adsorbed onto the surface of HKUST-1 by physical adsorption and interaction between drug molecules and the internal surface of HKUST-1, the addition of GO with HKUST-1 only provide a larger surface area, which gives less influence for drug release. As for the ibuprofen and ketoprofen release from MOFs and MOFs composite, the release rate of HKUST-1 composite is slower than that of HKUST-1. HKUST-1 composite exhibits a release of about 60% within 20 h, with complete release after 60 h. Non-linear regression was performed on the dissolution kinetics data using Equation (7).

Summary of drug release kinetic parameters and goodness of fit are summarized in Table 3. The reported R2 values show that Equation (7) is appropriate for describing the release kinetics of the 3 drugs from the different delivery vehicles. The dissolution rate constants for the release of felodipine from the MOF and MOF composite were similar. However, it can be seen from Table 3 that the drug release constants for the release of ibuprofen were significantly lower for the MOF composite case. This suggests that even though the inclusion of GO into the matrix increases the drug loading capacity, it retards the dissolution of the drug. This might be explained by the hydrogen bonding between carboxyl groups and hydroxyl groups of GO bonding with the carboxyl groups of ibuprofen and ketoprofen. The slower drug release profile of the ibuprofen drug from the HKUST-1 composite is likely due to the stronger adsorption of ibuprofen, as the GO leads to the bond between GO and ibuprofen.

Results show that at a temperature of 300.15 K, the solubilities of felodipine, ibuprofen, and ketoprofen were 352 mg/L, 763 mg/L, and 963 mg/L respectively, which improved a lot compared to 5 mg/L, 21 mg/L, and 160 mg/L, respectively, for the pure drugs. Also, it could be concluded that although the addition of GO might lead to the slow-release effect of ketoprofen and ibuprofen, the ability of drug loading could be significantly increased.

To evaluate the similarity or dissimilarity of the profiles, a similarity index used in the pharmaceutical industry to evaluate the similarity of dissolution profiles was adapted [43,44]. A similarity index was calculated from the grade efficiency curves from the two different profiles according to:(8)$$\chi =50log\left\{100{\left[1+\frac{1}{n}\sum_{t=1}^{n}{\left(f_{t}-f_{t,ref}\right)}^{2}\right]}^{-0.5}\right\}$$

In Equation (8), $f_{t,ref}$ is the amount dissolved (%) after time, *t*, *n* is the number of samples used in the drug dissolution curve; $f_{t}$ is the amount dissolved after time; *t* for the second-dissolution curve. The value of *χ* is between 0 and 100, and the value is 100 when the dissolution curves are identical. The values above 50 would indicate a profile similarity, whereas the values below 50 would indicate dissimilarity of the curves.

The calculated similarity values of the different HKUST-1 and composite materials are presented in Table 4. Here the comparison is between the dissolution profile of the matrix composed of HKUST-1 only and the composite material with 10% GO. The results show that the difference in the profiles increases in the order of felodipine < ketoprofen < ibuprofen. Based on the similarity values (*χ* > 50) reported in Table 4, the profiles of Ketoprofen and felodipine are not significantly different. However, for ibuprofen, the dissolution profiles are significantly different since *χ* < 50. The inclusion of graphene oxide in the matrix significantly affected the dissolution and release of ibuprofen.

## 4. Conclusions

In this work, MOFs-HKUST-1 and a new composite material consisting of model MOFs (HKUST-1) and graphite oxide were successfully synthesized and tested as a drug delivery vehicle. Three BCS Class II drugs ketoprofen, felodipine, and ibuprofen were used as the model drugs. The results show that the MOFs and MOFs-composite have shown great potential for use as drug delivery vehicles because the dissolution rate of the drugs from MOF -GO composite materials was significantly higher in comparison to the pure forms of the drug. The addition of GO to MOF provided more surface area onto the drug delivery matric and enhanced the drug loading, especially for ibuprofen and ketoprofen.

## Figures and Tables

**Figure 1 materials-16-01554-f001:**
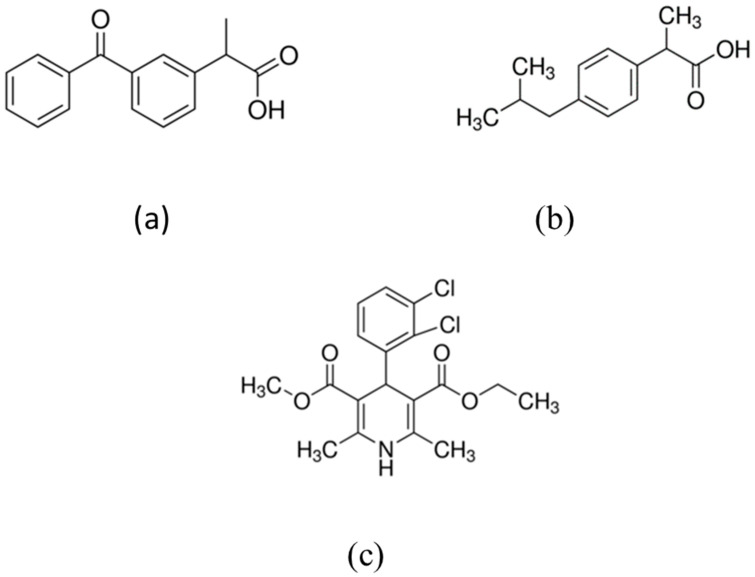
Chemical structures of (**a**) ketoprofen (**b**) ibuprofen and (**c**) felodipine.

**Figure 2 materials-16-01554-f002:**
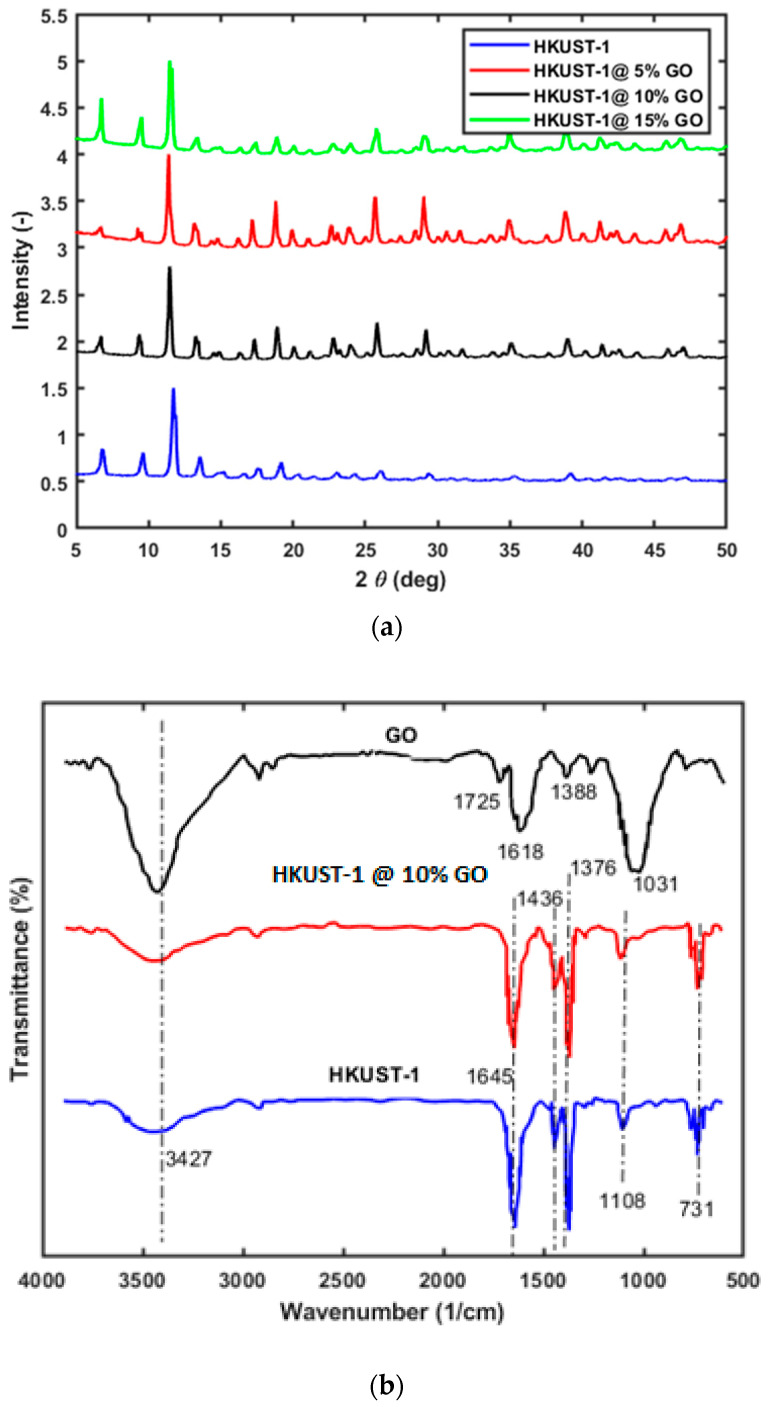
(**a**) Powder diffractograms of HKUST-1 and HKUST-1/GO composites. (**b**) FTIR spectra of GO, HKUST-1, and HKUST@ 10% GO samples.

**Figure 3 materials-16-01554-f003:**
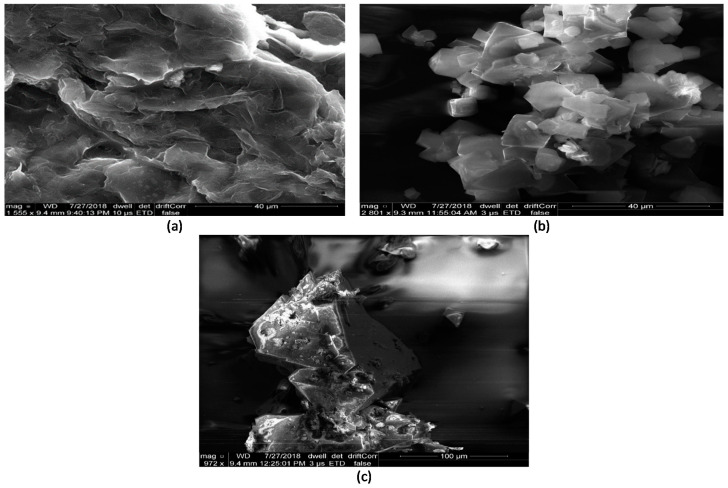
SEM images of powder samples (**a**) GO (**b**) HKUST-1 (**c**) HKUST-1 with 10% GO.

**Figure 4 materials-16-01554-f004:**
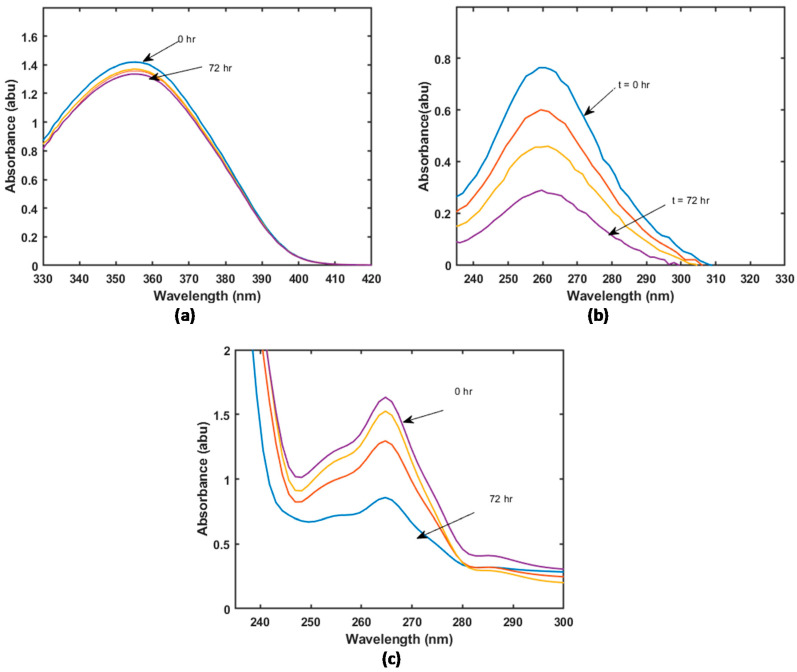
UV spectra of solution samples containing BCS Class II drugs taken at different times during loading of Ibuprofen onto HKUST-1. The initial concentration of the solution before loading MOF material was 1 g/L. (**a**) Felodipine (**b**) ketoprofen (**c**) Ibuprofen.

**Figure 5 materials-16-01554-f005:**
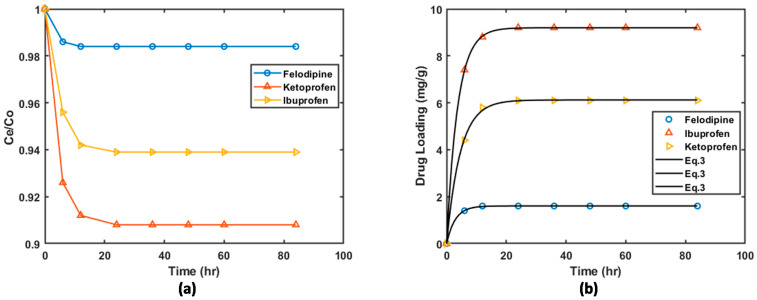
Drug loading kinetics of ibuprofen, ketoprofen, and felodipine onto graphene oxide (GO) (**a**) fraction of drug remaining of felodipine, ketoprofen, and ibuprofen onto GO (**b**) amount of drug load as function of time.

**Figure 6 materials-16-01554-f006:**
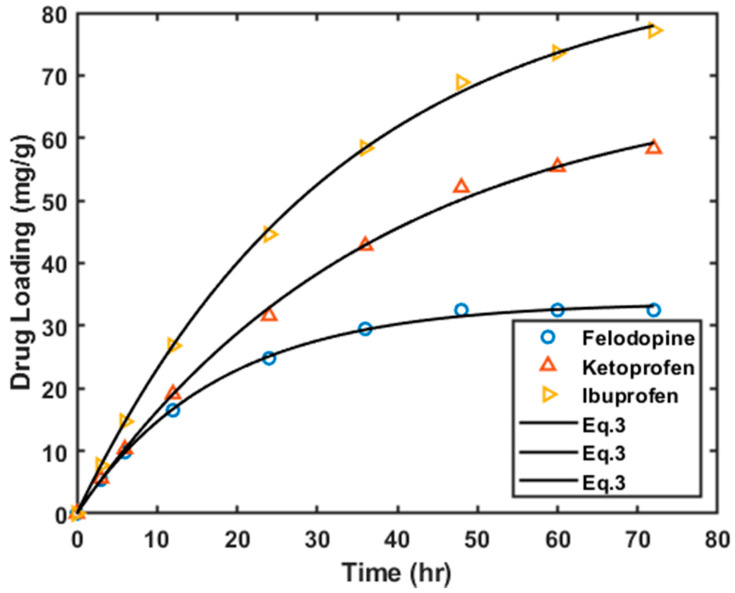
Loading kinetics of ibuprofen, ketoprofen and felodipine onto HKUST-1.

**Figure 7 materials-16-01554-f007:**
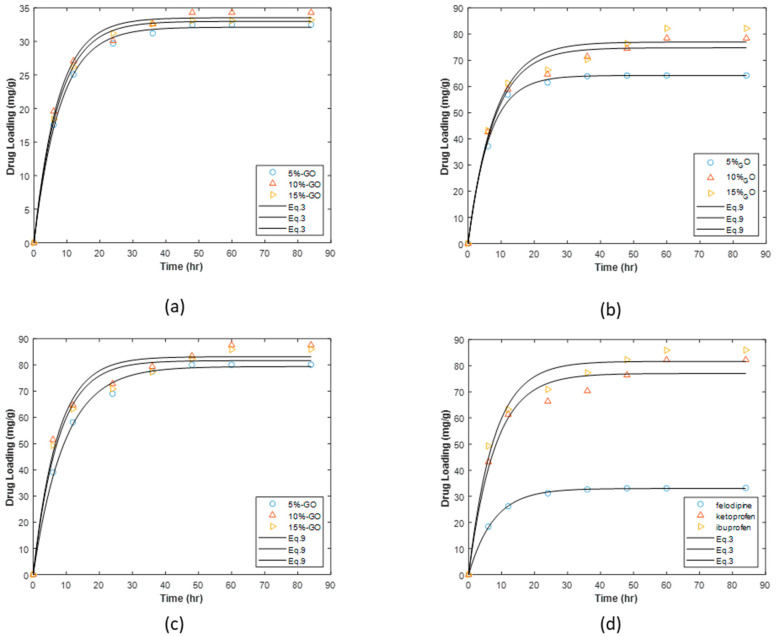
Loading kinetics of drugs onto HKUST-1/GO composite materials: (**a**) felodipine (**b**) ketoprofen (**c**) ibuprofen (**d**) comparison of the three drugs for the HKUST-10% GO composite.

**Figure 8 materials-16-01554-f008:**
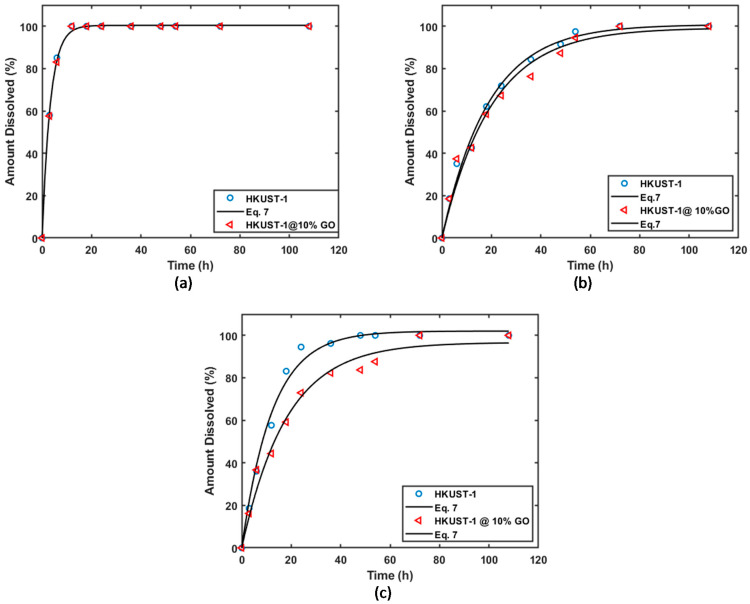
Drug release kinetics of drugs onto HKUST-1/GO composite materials: (**a**) felodipine (**b**) ketoprofen (**c**) ibuprofen.

**Table 1 materials-16-01554-t001:** BET results of GO, HKUST-1, HKUST-1 with 5%GO, HKUST-1 with 10% GO, HKUST-1 with 15%GO.

Material	BET Surface Area (m^2^/g)
GO	216
HKUST-1	1523
HKUST-1-5%GO	1612
HKUST-1-10%GO	1625
HKUST-1-15%GO	1608

**Table 2 materials-16-01554-t002:** Drug loading kinetic parameters for the different drugs onto MOF/GO composite materials with different amounts of GO.

Drug	Loading Vehicle	Loading Rate, *k_L_* (h^−1^)	*q_m_* (mg/g)	R^2^
Felodipine	HKUST-1	0.057 ± 0.013	33.7 ± 0.67	0.998
	HKUST-1@ 5% GO	0.127 ± 0.013	32.12 ± 0.67	0.998
	HKUST-1@ 10% GO	0.137 ± 0.027	33.53 ± 1.38	0.992
	HKUST-1@ 15% GO	0.133 ± 0.057	33.00 ± 0.30	0.996
	GO	0.349 ± 0.021	1.60 ± 0.01	0.999
Ketoprofen	HKUST-1	0.0346 ± 0.0033	58.30 ± 0.19	0.990
	HKUST-1@ 5% GO	0.156 ± 0.019	64.19 ± 1.57	0.997
	HKUST-1@ 10% GO	0.127 ± 0.036	74.76 ± 0.013	0.983
	HKUST-1@ 15% GO	0.126 ± 0.046	76.98 ± 5.92	0.976
	GO	0.271 ± 0.004	9.198 ± 0.021	0.999
Ibuprofen	HKUST-1	0.0220 ± 0.0012	79.10 ± 0.15	0.998
	HKUST-1@ 5% GO	0.107 ± 0.006	79.38 ± 2.53	0.996
	HKUST-1@ 10% GO	0.137 ± 0.044	83.14 ± 5.55	0.978
	HKUST-1@ 15% GO	0.132 ± 0.013	81.64 ± 5.52	0.978
	GO	0.218 ± 0.014	6.12 ± 0.07	0.9993

**Table 3 materials-16-01554-t003:** Summary of drug release kinetics parameter of different BSC II drugs from GO, MOF, and MOF—composites.

Drug	Loading Vehicle	Release Constant, $k_{R}\left(h^{-1}\right)$	R^2^
Felodipine	HKUST-1	0.2914	0.999
	HKUST-1@ 10% GO	0.2915	0.999
Ketoprofen	HKUST-1	0.0542	0.991
	HKUST-1@ 10% GO	0.0498	0.976
Ibuprofen	HKUST-1	0.081	0.999
	HKUST-1@ 10% GO	0.055	0.983

**Table 4 materials-16-01554-t004:** Comparison of the similarity factors between the release profiles from HKUST-1 and HKUST-1/10% GO composite.

Drug	Similarity Factor, χ
Felodipine	95.6
Ketoprofen	72.3
Ibuprofen	44.4

## Data Availability

Not applicable.

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
