# Peer review of "Enhancement of the Solubility of BS Class II Drugs with MOF and MOF/GO Composite Materials: Case Studies of Felodipine, Ketoprofen and Ibuprofen"

_materials, 2023, doi:10.3390/ma16041554_

Round 1

Reviewer 1 Report

The manuscript entitled as “Enhancement of the solubility of BS class II drugs with MOF and MOF/GO composite materials: case studies of felodipine, ketoprofen and ibuprofen” is interesting but not worthy enough to be accepted for publication in nanomaterials in its current form.

1.     Graphical abstract of whole procedure and reaction should be added.

2.     Abstract section has many grammatical errors and authors should also add numerical results of this research outcome.

3.     Abstract need to be improved with concise information of the research work and its results.

4.     There are mistakes of superscript and subscript.

5.     The grammatical and punctuation errors are also prominent in material and method and results part.

6.     The characterization results should be more elaborated in XRD and FTIR about the functional groups present in the compound.

7.     In figure 3, authors should mark/point out characteristic materials in SEM images. https://www.sciencedirect.com/science/article/pii/S1010603018302715, https://www.sciencedirect.com/science/article/pii/S0254058422006332

8.     Resolution of all figures is of poor quality so authors should improve it.

9.     The initial loading rate of the three drugs is also different.

10.  Page 11, line 275, authors check it carefully.

11.  Page 13, section “4. Discussion” should be “4. Conclusion”.

12.  The references used are from old papers. The references should be taken from latest studies and research paper. Authors should relevant article from present journal.

Author Response

The authors would like to express their gratitude for you valuable comments to improve the script. The comments have been addressed please see the attached document. 

Reviewer 2 Report

1.     Please mark the main peaks of the functional groups in Fig. 2b.

2.     The serious deficiency of the coordination of nano-HKUST-1-10% GO particles so as to make them easy to accumulate. Why?

3.     “Metal-organic frameworks (MOFs), a kind of porous hybrid inorganic-organic solids, have recently attracted great interest in many research areas such as catalysis, separation, gas-storage, and also drug delivery.” The sensor work may be also updated, such Micropor. Mesopor. Mat, 341(2022) 112098; Inorganics, 10(2022) 202; Inorg. Chem.,2015, 54, 6719-6726 and 2017, 19, 4368-4377.

4.     The BET results of HKUST-1-10%GO is so higher, why?

5.     Error! Reference source not found. What’s the matter?

6.     The figure quality of the Fig.7 and Fig. 8 should be improved.

7.     Why not study the releasing behavior of the loading samples?

Author Response

(The authors gave the same response as above.)

Round 2

Reviewer 1 Report

Authors have updated their draft according to the given suggestions do it maybe accepted in its current form.

Author Response

Thanks for the valuable comments. All suggested corrects have been made.

Reviewer 2 Report

accepted.

Author Response

Thanks for the valuable comments. The spelling errors have now been corrected.